# Surface networks in the Arctic may miss a future "methane bomb"

Sophie Wittig[1,*], Antoine Berchet[1], Isabelle Pison[1], Marielle Saunois[1], and Jean-Daniel Paris[1]

[1]Laboratoire des Sciences du Climat et de l'Environnement, CEA-CNRS-UVSQ, Gif-sur-Yvette, France
[*]Now at the Department of Meteorology and Geophysics, University of Vienna, Vienna, Austria

**Correspondence:** antoine.berchet@lsce.ipsl.fr

**Abstract.** The Arctic is warming up to four times faster than the global average, leading to significant environmental changes. Given the sensitivity of natural methane ($CH_4$) sources to environmental conditions, increasing Arctic temperatures are expected to lead to higher $CH_4$ emissions, particularly due to permafrost thaw and the exposure of organic matter. Some estimates therefore assume an Arctic "methane bomb" where vast $CH_4$ amounts are suddenly and rapidly released over several years. This study examines the ability of the in-situ observation network to detect such events in the Arctic, a generally poorly constrained region. Using the FLEXPART atmospheric transport model and varying $CH_4$ emission scenarios, we found that areas with a dense observation network could detect a "methane bomb" in 2 to 10 years. In contrast, regions with sparse coverage would need 10 to 30 years, with potential false positives in other areas.

## 1 Introduction

Arctic warming is proceeding three to four times faster than the global average. (AMAP, 2021; Rantanen et al., 2022). As a consequence, various environmental changes can be observed in high northern latitudes, triggering climate feedbacks that potentially accelerate global warming even further (AMAP, 2021). Those feedbacks include, for instance, increased greenhouse gas emissions (e.g. Treat et al., 2015), especially in the form of methane ($CH_4$). In the Arctic, $CH_4$ emissions are generally dominated by natural sources (e.g. Saunois et al., 2020; AMAP, 2015), including high northern latitude wetlands and other freshwater systems, fluxes from various oceanic sources, forest fires as well as geological fluxes. Quantifying natural $CH_4$ sources in the Arctic remains challenging and estimates are subject to large uncertainties. According to Saunois et al. (2020), wetland emissions above 60° N amount to 7 to 16 Tg $CH_4$ per year and other natural sources to 2 to 4 Tg $CH_4$ yr[-1]. However, as the Arctic region is not uniformly defined, comparing different estimates from various studies is an additional challenge. Since these natural $CH_4$ sources are sensitive to the surrounding environmental and climate conditions, it is assumed that $CH_4$ emissions will increase with progressing Arctic warming (e.g. AMAP, 2015).

This predicted increase is predominantly connected with permafrost thawing and the resulting exposure of large pools of degradable organic matter (Whiteman et al., 2013; Glikson, 2018). Regarding terrestrial permafrost, estimates predict that until 2100, up to 274 Pg of carbon could be released to the atmosphere, with $CH_4$ accounting for 40 to 70 % of the permafrost-affected radiative forcing (Schneider von Deimling et al., 2015; Walter Anthony et al., 2018). A potential increase in methane emissions from high northern latitude wetlands due to thawing permafrost soils has been indicated e.g. by Schuur et al. (2015).

Several studies have highlighted the importance of $CH_4$ emissions from the Arctic Ocean, particularly in shallow waters underlain by permafrost (Damm et al., 2010; Kort et al., 2012). Subsea permafrost thaw has been observed in the ESAS (East Siberian Arctic Shelf) and the importance of this region has been highlighted for instance by Shakhova et al. (2015, 2019) and Wild et al. (2018). Future estimates suggest that around 50 Gt of methane could be released from gas hydrates in the ESAS alone over the next 50 years (Shakhova et al., 2010), consistent with present annual estimates (e.g., Berchet et al., 2016).

Methane emissions from anthropogenic sources are estimated to be around at around 2 to 10 Tg $CH_4$ yr$^{-1}$ (Saunois et al., 2020). Anthropogenic $CH_4$ emissions in the Arctic are not explicitly assumed to increase in the future and several Arctic States report decreases in future emissions (Arctic-Council, 2019). However, the large estimates of unexplored fossil fuel resources make this region potentially attractive for future drilling campaigns (Gautier et al., 2009) and it has been confirmed that drilling has increased over the past decades in Arctic-boreal regions (Klotz et al., 2023).

The magnitude and multiplicity of possible climate feedbacks related to Arctic $CH_4$ natural emissions have been dramatically called *a sleeping giant*, (Mascarelli, 2009), *a methane time bomb*, (Glikson, 2018) or even *the methane apocalypse* (Ananthaswamy, 2015). However, different studies assessing an imminent Arctic "methane bomb" are more optimistic. McGuire et al. (2018) concluded that significant net carbon losses from northern permafrost regions will only occur after 2100, assuming effective climate action. Anisimov and Zimov (2021) demonstrated that $CH_4$ emissions from Siberian wetlands will increase by less than 20 Tg yr$^{-1}$ by 2050, leading to a global temperature increase of less than 0.02°C. Kretschmer et al. (2015) showed that $CH_4$ emissions from the ocean will remain limited over the next century despite significant losses of methane hydrates, particularly in the Arctic Ocean. Finally, Schuur et al. (2022) concluded that a sudden Arctic "methane bomb", releasing overwhelming amounts of $CH_4$ into the atmosphere in a short period of time, is not currently supported by observations or projections.

In Wittig et al. (2023), we used the existing network of atmospheric $CH_4$ concentrations in the Arctic in an inverse modelling system and concluded that no significant trend was observable in the last decade. Apart from the likelihood of an Arctic "methane bomb" in the near future, the objective of this study is to analyse the capability of a stationary observation network of atmospheric $CH_4$ concentrations to properly detect such a possible event in the future using atmospheric inversion. This is motivated by the general sparsity of the current (and planned) observation network in the Arctic. A 'methane bomb' is characterised in our study as a sudden and steep increase in methane emissions, releasing large amounts of $CH_4$ over several years. We focus hereby on the years 2020 to 2055. Consequently, this study aims to discuss the following questions: (*i*) could future increases of $CH_4$ emissions in the form of an Arctic "methane bomb" be accurately detected by the current observation network? and (*ii*) what improvements in the detectability of $CH_4$ emissions can be achieved by a hypothetically expanded network?

In order to implement this work, we apply hypothetical trend scenarios on different $CH_4$ emission sources to simulate a methane bomb in different regions located in high northern latitudes. By combining these emission scenarios with the extrapolated output of an atmospheric transport model, we obtain synthetic $CH_4$ mixing ratios for the current observation network in the Arctic and Sub-Arctic as well as for an observation network extended by possible additional sites. These synthetic observations subsequently serve as input data for the inverse modelling setup in order to identify a temporal threshold of possible

detection and to analyse regional differences in the ability of the two networks to adequately detect and localise increasing $CH_4$ emissions. Since we assume optimum quality and availability of the measurement data, the results obtained represent a best-case scenario for the detection of an Arctic methane bomb using exclusively in situ observations.

## 2 Synthetic-observation-based inversion method

Here, we implement an analytical inversion, aiming at explicitly and algebraically finding the optimal posterior state of a system $x^a$ and the corresponding uncertainties $\mathbf{P}^a$. This approach is defined by:

$$
\begin{cases}
\boldsymbol{x}^a & = \quad \boldsymbol{x}^b + \mathbf{K}(\boldsymbol{y}^o - \mathbf{H}\boldsymbol{x}^b) \\
\mathbf{P}^a & = \quad \mathbf{B} - \mathbf{KHB}
\end{cases}
\tag{1}
$$

with $\mathbf{K}$ the Kalman gain matrix given by:

$$
\mathbf{K} = \mathbf{BH}^T(\mathbf{R} + \mathbf{HBH}^T)^{-1}.
\tag{2}
$$

Our inversion system optimises $CH_4$ fluxes region-wise (121 regions, shown in Figure 1, Section 3.1) over a pan-Arctic domain, using atmospheric $CH_4$ concentrations. Our study examines scenarios spanning 36 years (2020-2055) to find a trade-off between computational cost and the importance of the decadal time scale for climate change. For computational reasons, this period has been split into 36 independent 1-year inversion windows, which are computed separately.

The prior knowledge of the state, in this case surface fluxes and soil uptake of $CH_4$, is defined by the control vector $x^b$ (see Section 3.3). Here, $x^b$ also contains information on the initial $CH_4$ background mixing ratios (described in Section 3.4), which are therefore optimised in addition to the $CH_4$ fluxes. The corresponding uncertainties are specified in the prior error covariance matrix $\mathbf{B}$. We use $\mathbf{B}$ matrices based on the Monte-Carlo log-likelihood approach developed in Wittig et al. (2023). The off-diagonal elements of the prior error covariance matrix are thereby determined by applying spatial and temporal correlations of 500 km and 7 days, respectively.

The observation operator is assumed to be linear since chemical oxidation of $CH_4$ by free radicals in the atmosphere is neglected for this application. It is therefore defined as its Jacobian matrix $\mathbf{H}$ and contains the simulated equivalents of the observations (further described in Section 3.4 and illustrated in Figure S2 in the supplements).

In classical inverse modelling approaches, the observation vector $\boldsymbol{y}^o$ contains available observations, e.g. on $CH_4$ mixing ratios. However, in this work we want to study different future scenarios of $CH_4$ emissions and therefore it is not possible to use actual measurements. Therefore, we simulate synthetic observations of $CH_4$ mixing ratios based on different emissions scenarios (further described in Section 3.5).

For a given emission scenario, the true state (hereafter called *truth*) of the $CH_4$ emissions over the future period of simulation is defined as $x^t$ and changes with a given trend $k$, which is constant throughout all the years within the period of interest. This

trend was only applied from the second year of the study period (2021), in the year 2020 the truth is identical to the prior state. The observations vector for a given year $j$ can then be calculated as:

$$
\begin{cases}
\boldsymbol{x}_j^t &= \boldsymbol{x}_{2020}^t \times (1+k)^{j-2020} \\
\boldsymbol{y}_j^o &= \mathbf{H}(\boldsymbol{x}_j^t).
\end{cases}
\tag{3}
$$

In our analysis of the detectability of elevated Arctic $CH_4$ emissions (Section 4), we examine how accurately the truth is captured in the posterior emissions of different regions and whether these elevated fluxes are localised in the right area. By design, our inverse modelling system will try to fit additional fluxes by adding $CH_4$ emissions in the Arctic region, but possibly not at the correct location. Since, as described above, the background mixing ratios are also included in the control vector $\boldsymbol{x}^b$ and consequently optimised in the posterior state, part of the missing $CH_4$ mass is likely to be compensated by increasing the background, hence generating a low-bias in the posterior emissions.

Similarly to the prior uncertainties, the matrix $\mathbf{R}$ containing the uncertainties on the synthetic observations as well as the modelled $CH_4$ mixing ratios is based on Wittig et al. (2023). Theoretically, the synthetic observations $\boldsymbol{y}_j^o$ should be perturbed by an error $\boldsymbol{\epsilon}_j^o$ (with a Gaussian distribution, following the matrix $\mathbf{R}$), accounting for measurement errors, as well as other uncertainties such as transport and aggregation (described e.g. by Szénási et al., 2021). In our approach, we deliberately disregard these errors in order to obtain optimistic results and assimilate optimal measurements to analyse the best possible detection of different observation networks (Section 3.2) regarding a methane bomb event.

## 3 Material

### 3.1 Region under study

For the implementation of the inversion, observation sites in high northern latitudes displaying different observation networks have been included in this study (see Section 3.2). To represent concentrations at these sites as accurately as possible, we simulate the influence of fluxes from a buffer region above 30° N. This region is subsequently divided into 121 sub-regions as proposed by the Regional Carbon Cycle Assessment and Processes (RECCAP; Ciais et al., 2022), in order to better detect local differences. Figure 1 shows the resulting sub-regions as well as all the included observation sites (indicated with white stars).

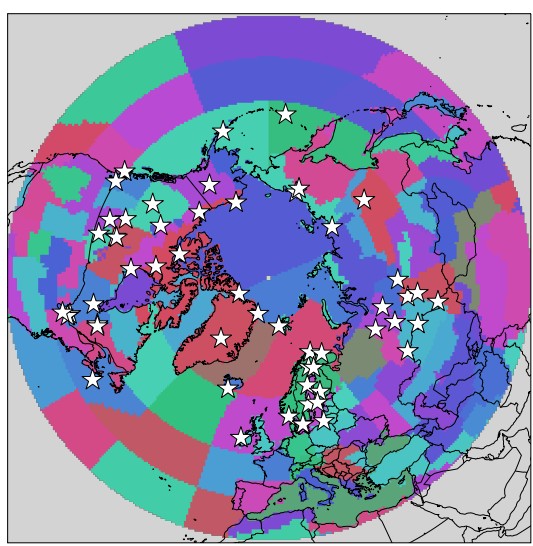

**Figure 1.** *RECCAP regions above 30° N. The white stars indicate all observation sites used in this study.*

### 3.2 Observation Networks

As described in Section 2, the observations used for the inverse modelling approach are based on synthetic $CH_4$ mixing ratios assuming different emission scenarios. We use, however, an existing network of measurement sites located in high northern latitudes. In this study, we focus exclusively on stationary $CH_4$ measurements, as our period of study spans several decades. Other types of greenhouse gas measurements, such as satellite observations, are currently limited to providing data for only a few years and are therefore not suitable for our purposes. The corresponding stations include both continuous and discrete measurements. To simulate an "optimal" observation network, we assume that all of those observation sites provide continuous measurements.

Two different network scenarios are used for this study. The first one, from here on referred to as current, includes all observation sites with available data of $CH_4$ mixing ratios during recent years. The term "current" refers hereby only to the

location of the stations. This network, as used in this study, already provides additional data compared to the actual observations available from these sites. This is because, as stated before, we assume continuous measurements where currently only discrete measurements are carried out. The current network contains hereby 40 stations in total, whereby the majority (26 sites) of the sites is located in North America (Canada, USA and Greenland). 10 observation sites are located in the Russian Arctic and

5 Sub-Arctic and 4 sites in Northern and Western Europe (Finland, Norway, Ireland and Iceland). The second network, referred to as *extended*, includes additional observation sites in high northern latitudes. The extended network expands the current network by 16 observation sites. The majority of these stations, 11, are located in Northern Europe (Sweden, Finland, Norway, Lithuania and East Russia), 3 in Central and Western Russia and one station each in Canada and Greenland.

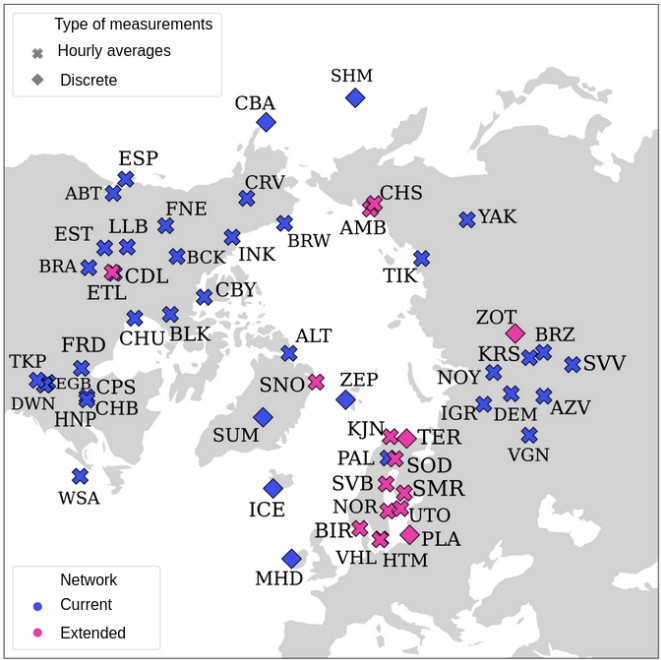

**Figure 2.** *Location of observation sites used to generate synthetic mixing ratio data. The current network is shown in blue, the additional stations in pink. Crosses indicate quasi-continuous, diamonds discrete measurements. The types of measurements refer to the measurements that are currently taking place at these sites, whereas in this study we assume all measurements to be continuous.*

Both the current and extended networks were selected based on their theoretical provision of $CH_4$ observations, including

10 measurements in the Russian Arctic that may currently not be accessible to the scientific communities of certain countries, as we believe it is important to conduct this work outside of ongoing political conflicts.

The different observation networks are shown in Figure 2 and an overview of both observation networks can be found in the Supplements in Table S1 (current network) and Table S2 (extended network).

The extended network hereby contains observation sites where measurements of atmospheric $CH_4$ concentrations are (*i*)

only available in recent years since 2022, (*ii*) not taking place anymore, or where (*iii*) the measurement data is not publicly

available, or where (*iv*) the stations use ground-based remote sensing instruments to obtain total column measurements of $CH_4$, or (*v*) $CH_4$ is currently not measured at all but measurements of other trace gases or air pollutants are taking place. As the observation network is limited at high northern latitudes, these additional stations were added to investigate what benefits a reasonably realistic extended network might offer for constraining methane fluxes.

## 3.3 Prior $CH_4$ emissions

The different methane sources and sinks used as prior information are based on a set of different emission inventories and land-surface models. Natural methane sources include hereby emissions from high-northern latitude wetlands, geological fluxes, $CH_4$ emissions from the Arctic Ocean, and wildfire events.

The $CH_4$ emissions related to anthropogenic activities include the exploitation and distribution of natural gas and mineral oil, agricultural activities as well as waste management and biofuel burning. Since anthropogenic activities are generally limited in the Arctic and Sub-Arctic, the corresponding datasets have been combined for simplification.

As mentioned before, atmospheric $CH_4$ sinks from free radicals are not taken into account. However, soil oxidation due to microbial activities is included in the form of negative $CH_4$ emissions. All prior estimates are listed in Table 1

**Table 1.** *Methane sources and sink taken into account in the prior emissions.*

| Type | Source | Reference | Temporal resolution |
|------|--------|-----------|---------------------|
| Natural | Wetlands | Poulter et al., 2017 | monthly climatology |
| | Ocean | Weber et al., 2019 | constant |
| | Geological | Etiope et al., 2019 | constant |
| | Soil Oxidation | Ridgewell et al., 1999 | monthly climatology |
| Combined | Biomass and | GFED4.1 | monthly with |
| | biofuel burning | EDGARv6 | interannual variability |
| Anthropogenic | Mineral oil & gas | EDGARv6 | interannual variability |
| | Waste & Agriculture | EDGARv6 | interannual variability |

## 3.4 Synthetically generated $CH_4$ mixing ratio data

The modelled $CH_4$ mixing ratios are obtained by simulating backward trajectories of virtual particles using the Lagrangian atmospheric transport model FLEXPART (FLEXible PARTicle) version 10.3 (Stohl et al., 2005; Pisso et al., 2019).

In this study, 2000 particles are released once per day at each observation site (Section 3.2) and followed 10 days backwards in time. The horizontal resolution is hereby $1° \times 1°$. The meteorological input data for the FLEXPART simulations is provided by the European Centre for Medium-range Weather Forecast (ECMWF) ERA5 (Hittmeir et al., 2018) with 3-hourly intervals and 60 vertical layers.

The so-called footprints obtained by sampling the near-surface residence time of the various backward trajectories of the virtual particles are subsequently used to determine the $CH_4$ mixing ratios per methane emission sector (Section 3.3) and sub-region (Section 3.1). The footprints define hereby the connection between the methane fluxes discretised in space and time and the change in concentrations at the observation site (Seibert and Frank, 2004). To obtain a time series of modelled $CH_4$ mixing ratios, a time series of footprints is integrated with discretised $CH_4$ flux estimates.

As described in Section 2, in the inverse modelling framework, the modelled $CH_4$ mixing ratios obtained from the FLEX-PART footprints are included in the observation operator $\mathbf{H}$. In this study, this matrix is used for both the calculation of the synthetic future observations (shown in Equation 3) based on future emission scenarios (see Section 3.5) as well as their modelled equivalents based on prior emission estimates.

Since the thus obtained $CH_4$ mixing ratios only display short-term fluctuations at the observation sites, the background mixing ratios need to be taken into account. Those are hereby calculated by combining a $CH_4$ concentration field as initial condition with the FLEXPART backward simulations (e.g. Thompson and Stohl, 2014; Pisso et al., 2019). The initial concentration field is provided by the Copernicus Atmospheric Monitoring Service (CAMS): a $CH_4$ mixing ratio field from CAMS global reanalysis EAC4 (ECMWF Atmospheric Composition Reanalysis 4) with 60 vertical layers, a 3-hourly temporal and a $0.75° \times 0.75°$ spatial resolution has been used (Inness et al., 2019). The implementation used for obtaining the background mixing ratios is provided by the Community Inversion Framework (CIF; Berchet et al., 2021). However, since an exact estimate of the background mixing ratios remains challenging and the calculated background concentrations do not provide perfect estimates, the background mixing ratios are optimised together with the $CH_4$ fluxes (see Section 2).

As mentioned in Section 3.1, the period under study covers the years 2020 to 2055. To represent this period of time, which partly lies in the future, we use FLEXPART simulations covering the 12 years (between 2008 and 2019) and string together this sequence of footprints three times in a row. It is hereby assumed that the climatology of atmospheric transport and fluxes does not change significantly in the 36 years following the year 2019.

### 3.5 Future emission scenarios

We create various scenarios by varying four different parameters: (*i*) $CH_4$ emission sources, (*ii*) the trend on these sources, (*iii*) the regions in which the trends are applied, (*iv*) the observation network.

Hypothetical trends are applied, in varying regions, to wetlands, anthropogenic activities, and the Arctic Ocean (Table 2). We define five supra-regions (see Supplements, Figure S1): the Arctic, the Arctic and Sub-Arctic combined (hereafter named *entire region*), North America, East Eurasia, and West Eurasia; the last three regions only refer to high northern latitude areas within those continents. Additionally, 121 sub-regions are defined as detailed in Section 3.1. In total, the trends are therefore applied to 126 different regions including both the sub- and the supra-regions.

For each of these zones, positive trends are applied separately on wetland and anthropogenic emissions. Oceanic $CH_4$ emissions are only increased in the sub-region that contains the ESAS, as these are difficult to detect with the surface networks.

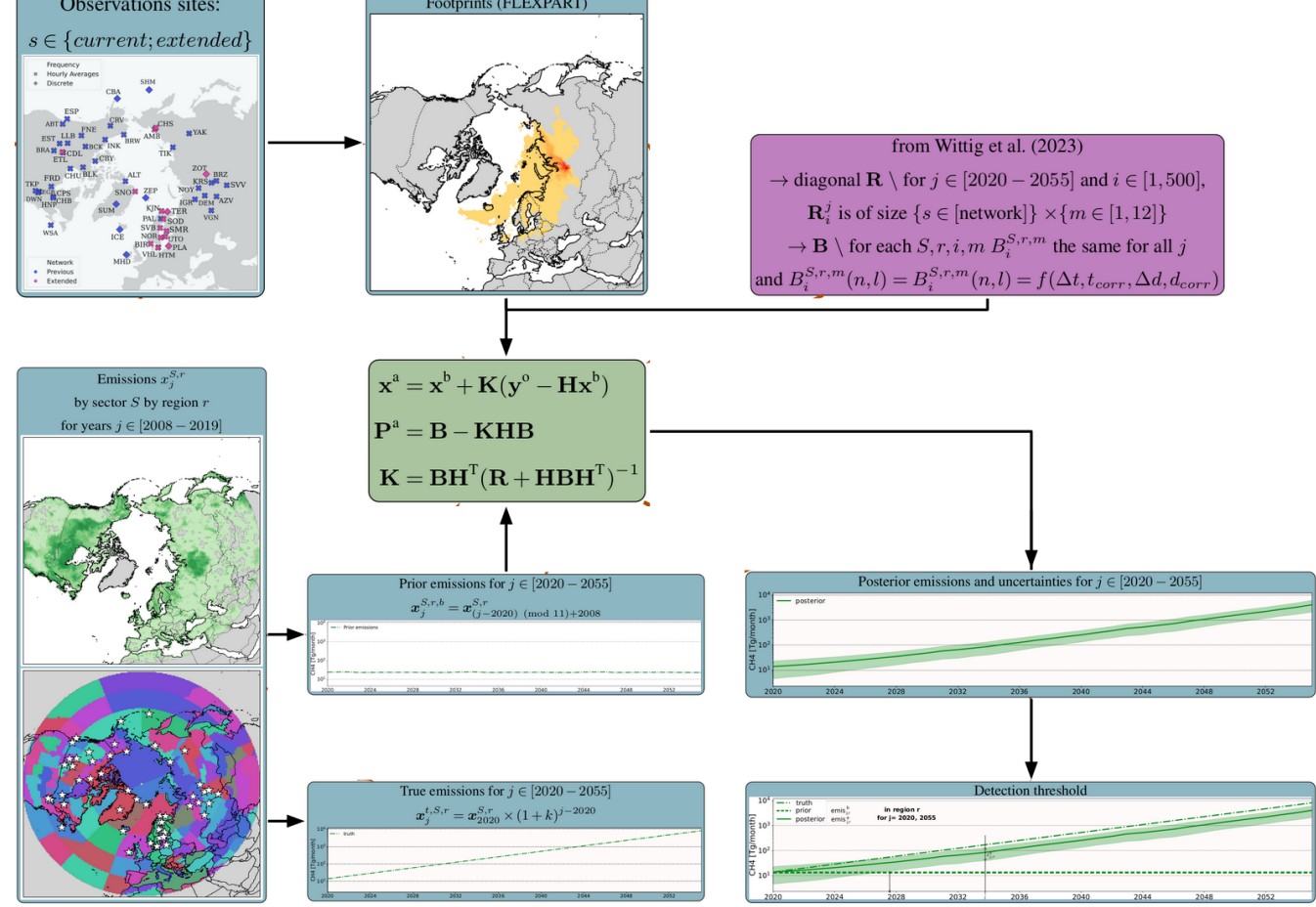

**Figure 3.** *Principle of the inversion set-up used in this study. The modelled input and output data of the inversion are shown in the blue boxes, the respective uncertainties in the purple box and the optimisation strategy in the green box. See Section 2 and Wittig et al. (2023) for full details.*

The trends are hereby varied between a 0.1 and 20% increase per year for anthropogenic and wetland emissions and between 1 and $100\%\,\mathrm{y}^{-1}$ for oceanic sources. As the results obtained are applicable to both lower and higher trend scenarios, we focus only on the highest selected increase (20% for wetlands and anthropogenic sources and 100% for oceanic fluxes), as this is also the most representative for a "methane bomb" event.

5    For both anthropogenic and wetland emissions we obtain 252 separate scenarios when increasing the emissions in each of the 126 regions and using the two different observation networks. Since oceanic fluxes are only increased in one region, we obtain only two scenarios using the different observation networks. This results in 506 different setups with corresponding synthetic observations. Hence, the same number of inversions are carried out. The main elements for the ensemble of inversions run in this study are summarized in Figure 3 and detailed in Section 2.

**Table 2.** *The different scenarios providing the simulated observations.*

| Methane Source | Region | Trend [% per year] | Network |
|---|---|---|---|
| Wetlands | All 5 supra-regions, All 121 sub-regions | 20 | current, extended |
| Anthropogenic | All 5 supra-regions, All 121 sub-regions | 20 | current, extended |
| Ocean | Only ESAS region | 100 | current, extended |

## 4 Results

Section 4.1 illustrates how the true and the posterior fluxes evolve over time in the Arctic for one selected scenario. We evaluate the performance of the inversion not only through how close to the true fluxes the posterior fluxes get but also by the time at which a trend appears in the posterior fluxes compared to the flat prior. This is described in the Section 4.2.

### 4.1 Comparison of truth and posterior state over time

In order to evaluate how well the anticipated trends in the different regions are captured over the whole period of interest, the time series of the true and posterior states are compared to each other. The true state refers hereby to the emission scenario used to compute the synthetic observations. Figure 4 shows the time series of wetland and total $CH_4$ emissions between the years 2020 and 2055 in different supra-regions (North America, East Eurasia, and the Arctic) as well as the total $CH_4$ emissions for the entire regions. The truth is hereby a 20 % increase in wetland emissions only in the supra-region East Eurasia and the extended observation network was used for the inversion.

Since wetland emissions are only increased in East Eurasia in this scenario, only this region should be updated by the inversion. It is shown that the posterior emissions are indeed increasing in this region, however, at a lower rate than intended by the scenario. By the year 2055, the posterior emissions ($\approx 4092$ Tg $CH_4$ yr$^{-1}$) are approximately 50 % lower than the truth ($\approx 8152$ Tg $CH_4$ yr$^{-1}$). This is also found for the total emissions in the entire Arctic and Sub-Arctic region, where the posterior emissions are around 28 % lower than the truth in 2055.

However, it is shown that the posterior wetland emissions are also increasing over time in regions where no trend was applied, such as North America. Here, the posterior state starts deviating from the truth since around 2032. At the end of the period in 2055, the annual $CH_4$ emissions from wetlands are $\approx 330$ Tg higher than the given unmodified truth ($\approx 30$ Tg $CH_4$ yr$^{-1}$). This means that the increase retrieved in the posterior state is underestimated compared to the generated truth in the "correct" area, which is considered to be the true state of the emissions in this inversion set-up. This is partially compensated for in the total posterior by overestimations in the same emission sector in different regions.

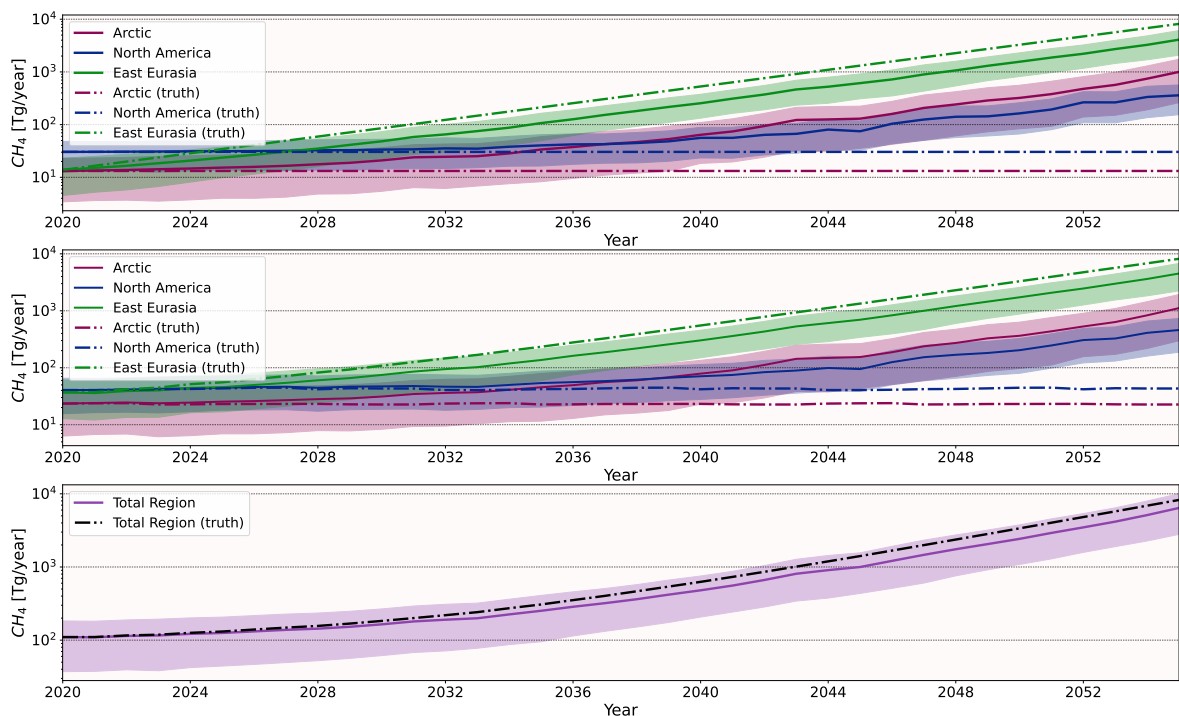

**Figure 4.** *Time series of emissions [TgCH₄ yr⁻¹] between 2020 and 2055 with a 20%-per-year increase in wetland emissions in East Eurasia. The continuous lines show the posterior state, and the dash-dotted the true state. The Arctic is shown in pink, North America in blue, East Eurasia in green, and the entire region in purple. The shaded areas refer to the posterior uncertainties obtained from the $\mathbf{P}^a$ matrix. Top panel: regional wetland CH₄ emissions, middle panel: regional total CH₄ emissions, bottom panel: entire total CH₄ emissions.*

When the same emission scenario (20 % increase in wetland emissions) is applied exclusively to North America, the opposite effect is observed: the posterior emissions in North America are underestimated to be around 26 % lower than the truth, and in East Eurasia the posterior CH₄ fluxes are significantly higher compared to the truth. The discrepancies are, however, lower in comparison to the scenario anticipating elevated wetland emissions in East Eurasia. This can be explained by the denser observation network available in North America, resulting in a better posterior distribution of fluxes. Similar results are obtained under elevated anthropogenic CH₄ emissions.

## 4.2 Regional trend detection

Subsequently, we analyze how well the prescribed trends in the different regions are detected by the inversion in the posterior state. In order to summarise the results of the numerous scenarios, all the figures presented in this section encompass 121 scenarios described in Section 3.5: in 120 of these scenarios, the trend was applied on wetland emissions in only one of the corresponding sub-regions, in the remaining scenario the trend was only applied to oceanic CH₄ emissions in the ESAS region

(see Supplements, Figure S1d). These scenarios are chosen for the illustration figures since similar results are obtained for anthropogenic CH$_4$ emissions.

### 4.2.1 Trend detection threshold

We define a temporal threshold in each of the 121 sub-regions $r$ in order to determine when the posterior state is statistically different from the prior.

For this, we select the years for which the difference between the annual posterior emissions in year $j$ and region $r$ $emis^a_{j,r}$ and the prior $emis^b_{2020,r}$ is larger than the absolute posterior error $\epsilon^a_{j,r}$ in the threshold year:

$$emis^a_{j,r} - emis^b_{2020,r} < \epsilon^a_{j,r} \tag{4}$$

with $j \in [2021, 2055]$ and $r \in [1, 121]$,

The threshold year is therefore defined as the first year, for which equation 4 is **not** fulfilled.

Due to the looping of footprints and fluxes from 12 years to generate the future truth (described in Section 2), the criterion may be matched for some years discontinuously first, then continuously until 2055. The threshold is therefore the first year after which all years are flagged as detected, as illustrated in Figure 3. As expected, the threshold year is generally later for regions with a sparse observation network (Figure 5a).

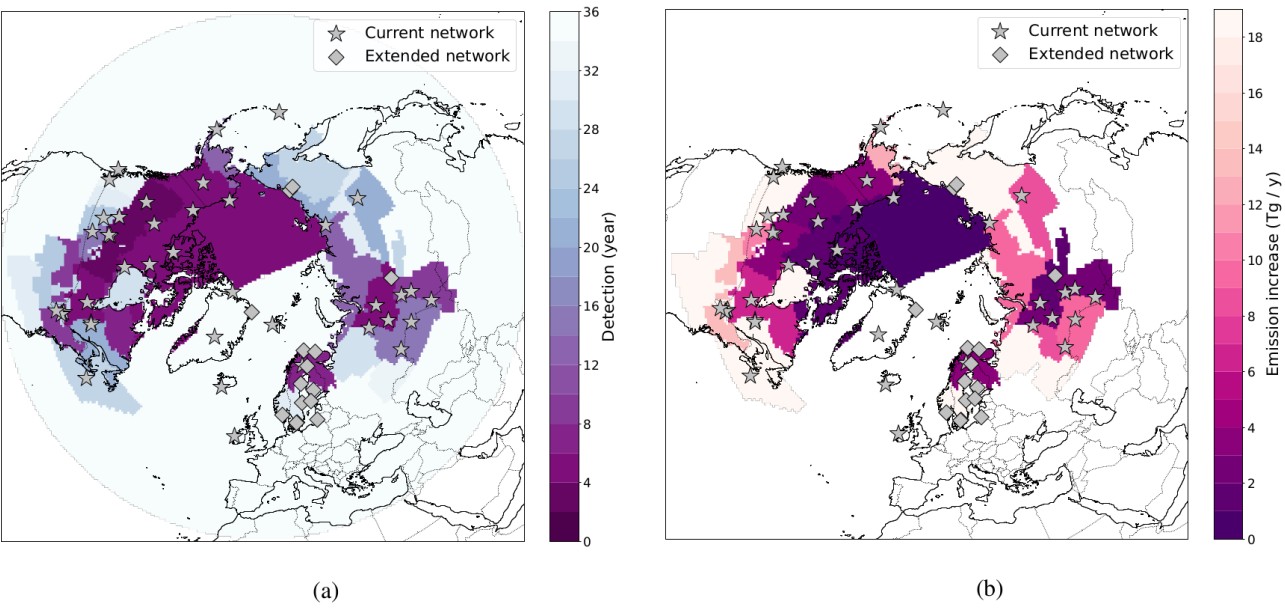

(a)  (b)

**Figure 5.** *(a) Threshold year counted from 2020 for each sub-region. In the ESAS region, the trend applied to ocean emissions is 100 % y$^{-1}$; for all other regions, a trend of 20 % y$^{-1}$ is applied to wetland emissions. The inversion is performed using the current observation network only (grey stars). The stations of the extended network are indicated by grey diamonds. (b) Increment in yearly emissions for each sub-region at the threshold of detection, in Tg yr$^{-1}$.*

In regions with a dense network, such as northern North America, the threshold year is quite early (after $\approx 5$ years over 36). These figures reflect an ideal case where uncertainties in the inversion system are minimised, in particular on the synthetic observations as described in Section 2, and it is assumed that data are immediately available. In reality, it could take much longer to detect a significant trend, even in regions with relatively dense networks. Moreover, the applied trend of $20\%\mathrm{y}^{-1}$ for wetlands and $100\%\mathrm{y}^{-1}$ for ESAS is particularly pessimistic. For example, a trend of $+20\%\mathrm{y}^{-1}$ for wetlands in East Eurasia results in emissions increasing from less than 14 Tg in 2020 up to 8150 Tg in 2055, totally unrealistic compared to present day global emissions of 550-880 Tg yr$^{-1}$ (Saunois et al., 2020). Hence, it is more illustrative to analyse the smallest amount of emissions which can be detected, as shown in Figure 5b, than simply using the year of detection as an indicator. As also observed for the threshold year, the emission threshold is generally smaller near the denser part of the network. In most regions, even in the most favourable parts of the Arctic in terms of detection limits, an increase of a few, up to 10 Tg yr$^{-1}$ (which corresponds to an increase of approximately 7 % per year), is necessary for statistically reliable detection. Such detection thresholds are close to the expected emission increases in the coming decades, e.g., 20 Tg yr$^{-1}$ from thawing permafrost in Siberia (Anisimov and Zimov, 2021). This raises possible limitations in the detection of such events, as the detection limits further away from the observation networks are much higher. More realistic scenarios would take much longer to be detected.

### 4.2.2 Detection of trend magnitudes

Subsequently, we want to examine how well the previously determined trends of 20 % increase in wetland emissions and 100 % increase in oceanic CH$_4$ emissions, respectively, are captured in each of the corresponding sub-regions.

Therefore, the relative difference $\Delta emis_{j,r}$ is the difference between the posterior annual CH$_4$ emissions $emis^a_{j,r}$ in the threshold year defined in Section 4.2.1 and the corresponding truth $emis^t_{j,r}$ divided by the truth in the threshold year:

$$\Delta emis_{j,r} = \frac{emis^a_{j,r} - emis^t_{j,r}}{emis^t_{j,r}} \tag{5}$$

for $j \in [2021, 2055]$ and $r \in [1, 121]$. Therefore, the closer the difference $\Delta emis_{j,r}$ is to zero, the better the truth is captured in the posterior state of the corresponding sub-region.

As expected, the posterior increment in the defined threshold year is closer to the truth in areas with a dense observation network (Figure 6a). Those include North America, parts of Siberia, the RECCAP region containing ESAS and parts of Northern Europe: The posterior results deviate from the truth approximately between 0 and 45 %. The exceptions are some oceanic regions outside the Arctic Ocean. Here, the small differences between the posterior emissions and the truth are unrelated to the observation network, but due to the absence of trends.

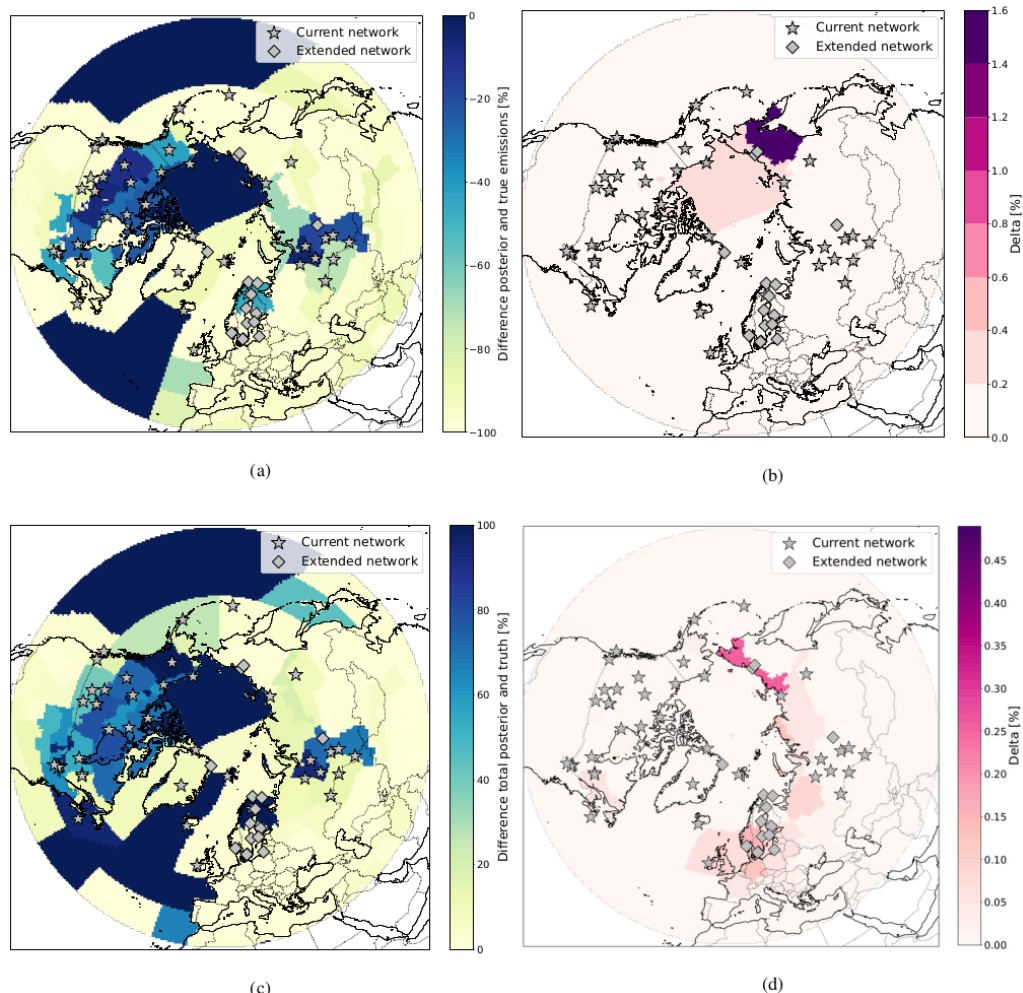

**Figure 6.** *(a) Relative difference [%] between posterior and true annual CH₄ emissions [Tg CH₄ yr⁻¹] in the threshold year of the corresponding region. Darker shades indicate regions where the increment in the posterior state is closer to the truth. The inversion is performed using the extended network. (b) Difference between current and extended observation networks regarding the relative differences between the truth and the posterior state. (c) Ratio of total CH₄ emission in pan-Arctic domain and true increment corresponding sub-region. (d) Difference between current and extended observation network regarding the total and true increment.*

Additionally, in order to determine the share of the truth detected by the inversion, we calculate the detection ratio $K_{j,r}$. Hereby, the posterior increment in all regions $\Sigma \Delta emis^a_{j,r}$ in the threshold year $j$ is divided by the the true increment $\Delta emis^t_{j,r}$ in region $r$:

$$K_{j,r} = \frac{\sum \Delta emis^a_{j,r}}{\Delta emis^t_{j,r}} \tag{6}$$

with $j \in [2021, 2055]$ and $r \in [1, 121]$.

Hence, we analyse how much of the true increase is detected, independent from the location it is attributed to, when increasing the CH$_4$ emissions in one of the sub-regions. Higher values indicate that a larger share of the true emissions is detected in the posterior emission, distributed over the whole pan-Arctic domain. Figure 6c shows that the detection ratio is generally higher when the true emissions are increased in regions with a dense observation network (such as North America), with values of up to 100 %. Similar to the relative difference (Figure 6a), the high detection ratios in the oceanic regions are due to the absence of trends in the true emissions, since the CH$_4$ emissions in these regions are nearly zero.

When comparing the two observation networks, the improvements achieved by the additional sites are remarkably small: the posterior state of the extended network is closer to the truth by a maximum of 1.6 % in comparison to the current network (Figure 6b). Regarding the comparison of the detection ratio of the two networks, shown in Figure 6d, the improvement is even smaller with a maximum of 0.3 %. Only the two added stations at the coast of the East Siberian Sea (AMB and CHS) seem to provide additional constraints for the surrounding regions. One possible reason for this could be related to the locations of the additional observation sites, as several of them are located close to operating measurement stations and/or in areas with low estimated CH$_4$ fluxes.

In Northern Europe, where the network was extended by 10 sites, the differences between the current and the extended networks are not significant. This is related to our particular set-up, for which background concentrations are optimised alongside fluxes. In Northern Europe, despite the provision of numerous additional sites, the inversion attributes observation discrepancies between the truth and the prior to the background concentrations instead of the fluxes.

### 4.2.3 Misattribution of CH$_4$ emissions

The inversion may produce artefacts and "detect" trends not only in the region where a trend is applied to the truth but also in other regions. To assess this issue, we calculate how much increase is detected in the posterior CH$_4$ emissions in all other regions for the given threshold year in relation to the growth detected in the region in which the increment is actually applied. In other words, we evaluate how much emissions due to the trend in the region examined is "misattributed" to other regions.

For instance, for an applied trend in RECCAP region $i$, the posterior increment ratio $\kappa_{j,i}^a$ can be defined as:

$$\kappa_{j,i}^a = \frac{\sum \Delta emis_{j,r}^a}{\Delta emis_{j,i}^a} \tag{7}$$

for the threshold year $j \in [2021, 2055]$ and the region $r \in [1, 121]$ $r \neq i$. $\Delta emis_{j,r}^a$ and $\Delta emis_{j,i}^a$ hereby represent the difference between the posterior CH$_4$ emissions in the threshold year $j$ and the true emissions in the year 2020 in the corresponding region $r$ or $i$, respectively.

Areas with a denser observation network generally show less misattribution of CH$_4$ fluxes to other regions (Figure 7a), following the results presented in Section 4.2.2. Hereby, the posterior increment ratio in other regions is around 0 to 40 %. For areas with a sparse network of surface observation sites, increases in CH$_4$ fluxes in other regions can be more than 1000 %.

The improvements achieved through the expansion of the network are more substantial regarding the misattribution of CH$_4$ fluxes to other regions (Figure 7b), compared to the results presented in Section 4.2.2. For example, the improvement by the

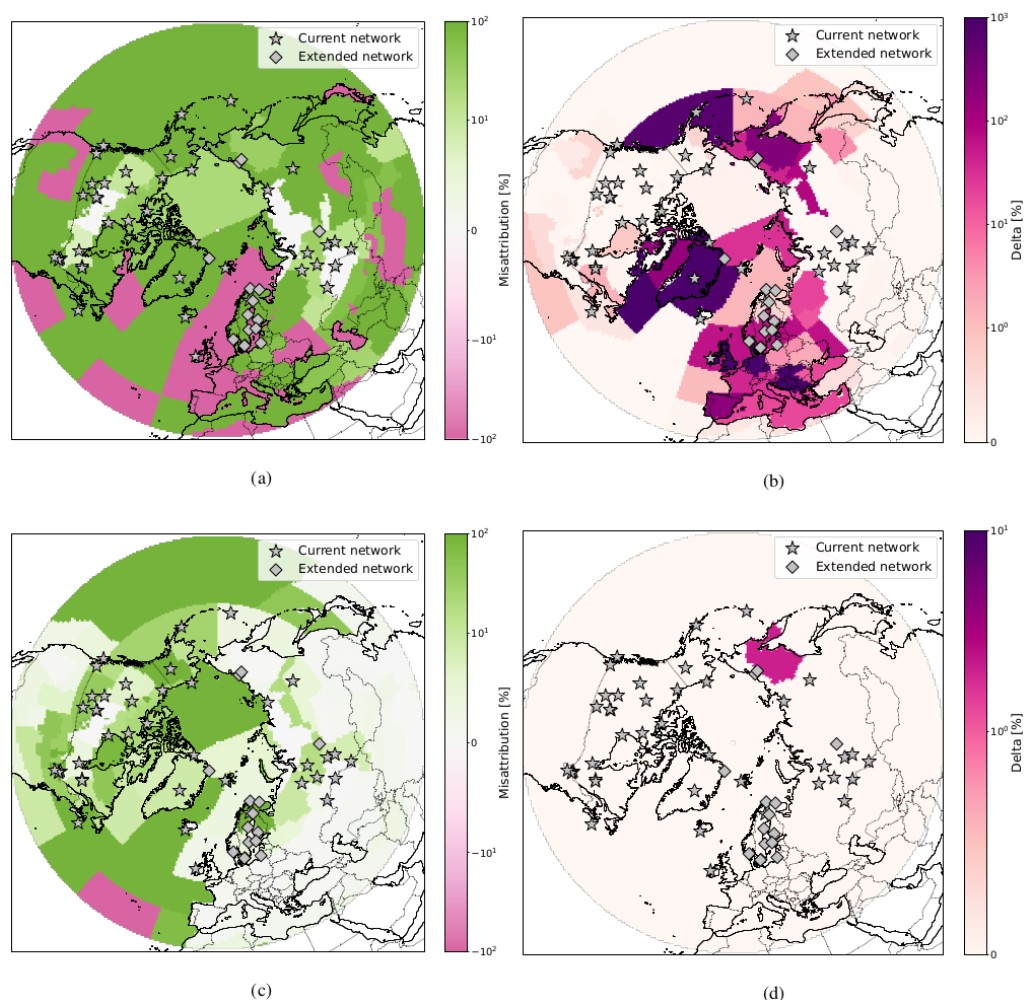

**Figure 7.** *(a) Misattribution of detected CH₄ emissions to regions other than the region a trend is applied to. Deeper shades show hereby a large increase (green) or decrease (pink) in other regions. Areas coloured in deep green show regions for which the increment outside is much larger than inside the region, where the increment was intended; pink coloured regions tend to decrease CH₄ fluxes outside. The closer to white the colour of a region, the less the emissions are modified outside of it. (b) Difference between the posterior increment ratio of current and extended network. Darker shades of purple show regions where the extended network performs better in comparison to the current network regarding the misattribution. (c) Misattribution of true CH₄ emissions (colours as in Figure (a)). (d) Difference of true increment ratio between current and extended observation network.*

two stations, AMB and CHS, described in the previous section can also be observed here. For the region those sites are located in, the posterior increment ratio was 286 % in the scenario using the current network and only 34 % in the extended network. Improvements are also found in Europe and Greenland.

In addition to the posterior increment ratio, we compute the true increment ratio $\kappa_{j,i}^t$ for each sub-region $i$:

$$5 \quad \kappa_{j,i}^t = \frac{\sum \Delta emis_{j,r}^a}{\Delta emis_{j,i}^t} \tag{8}$$

for the threshold year $j \in [2021, 2055]$ and the region $r \in [1, 121]\ r \neq i$. $\Delta emis_{j,i}^t$ is hereby defined as the difference between the true $CH_4$ fluxes in the threshold year $j$ and the truth in 2020 in the corresponding region $i$. The closer the value of $\kappa_{j,i}^t$ of a specific region is to zero, the less true emissions are misattributed to other sub-regions. The true increment ratios are shown in Figure 7c. Similar to the posterior increment ratios, the fluxes are generally less misattributed when the true emissions are

10 increased in continental areas with available observation sites, especially in Siberia and Canada. The improvements from the extended observation network are smaller regarding the true increment ratio (see Figure 7d) in comparison to the posterior increment ratio, with only one region in eastern Siberia showing a clear improvement of around 10 %.

## 5  Conclusions

In this study, we generated numerous future scenarios simulating an assumed "methane bomb" in high northern latitudes. To

15 determine how well the existing in situ observation network (consisting of 40 sites) as well as a possible future network (56 sites) are able to detect increases in $CH_4$ emissions, those scenarios were integrated in an analytical inversion framework.

The period under study covers the years 2020 to 2055. During this period, different annual increases are applied to three $CH_4$ sources: wetlands, oceanic sources and anthropogenic emissions. Those scenarios of possible trends were applied to different sub-regions in the high northern latitudes. The particular "methane bombs" due to each type of source are not discussed

separately here. In fact, it is likely that emissions from these $CH_4$ sources will increase simultaneously as a result of Arctic warming. Therefore, we focus on spatial patterns in order to detect trends.

In this approach, we have made the optimistic assumption of excellent quality and availability of measurement data. The results presented therefore represent the best possible scenario for detecting a future Arctic methane bomb.

The posterior $CH_4$ emissions are underestimated (by up to 41 %) in most regions a trend was assigned to. The discrepancies

are larger in later years and proportional to the magnitude of the true trend. Additionally, increasing posterior $CH_4$ fluxes are also found in regions where increasing emissions are not prescribed. This effect is smaller when the true trend is assigned to regions with a dense observation network. However, the additional hypothetical sites bring little improvement in this regard. This indicates that neither of the two observation networks is able to correctly quantify and locate increases in Arctic methane emissions.

For the correct detection of the true trend in a specific area, the regional differences confirm that detection is better in regions with numerous observation sites, such as northern North America or parts of Siberia. Still, the improvements achieved by the

extended observation network are remarkably small. A noticeable improvement is only found in the north-east of Russia, and the detection is only up to 1.6 % better than with the current network.

A more significant advantage of the extended observation network is linked to the misattribution of $CH_4$ fluxes. As stated before, the results show that increased $CH_4$ emissions are not only detected in the region where the trend actually occurs, but that "false positives" are detected in other regions. The inversion set-ups using the extended observation network show significant improvements, for instance in the north-east of Russia, Europe, and Greenland.

Overall, this study shows that "methane bombs" could be detected in Arctic regions with good observational coverage within 2 to 10 years, while in poorly covered regions detection would take 10 to 30 years, with the added risk of triggering false positives in other regions.

Therefore, efforts to integrate mobile campaigns and new-generation satellite observations into inverse modelling systems should be supported and developed further. Satellite observations in particular offer a high potential to compensate for the lack of in situ observations in the Arctic. The feasibility of using available satellite data products for inverse modelling of methane emissions in high northern latitudes was, for instance, discussed by Berchet et al. (2015) and several approaches integrating these observations in Arctic regions (e.g., TROPOMI $CH_4$ products, Tsuruta et al., 2023) have been implemented. However, the quality of the data provided by currently operating remote sensing instruments is hampered in high northern latitudes by factors such as high solar zenith angles, low albedo of the Arctic Ocean and limited daylight during polar nights. However, new satellite missions (e.g., the Franco-German MERLIN project) will possibly provide large, accurate and high-resolution data sets, suitable for characterising $CH_4$ plumes from regional sources and better constraining methane fluxes in the Arctic.

Current political differences as well as the associated sanctions are an additional obstacle regarding the accessibility of crucial $CH_4$ observations in the Russian Arctic and Sub-Arctic. As the network in this region is already limited, this missing information may further hamper obtaining a complete picture of ongoing processes in the Arctic, including the detection of a possible methane bomb.

*Code and data availability.* FLEXPART is an open-source model and can be downloaded here: flexpart.eu. The meteorological forcing fields are interpolated from open ERA5 re-analysis, extracted using the open-source flex-extract toolbox (Tipka et al., 2020, flexpart.eu/flex_ extract; last access: 01/10/2023). Flux data were obtained from the Global Carbon Project - $CH_4$ (icos-cp.eu/GCP-CH4/2019; last access: 01/10/2023). The background concentrations were calculated using the Community Inversion Framework: community-inversion.eu, Berchet et al. (2021).

*Author contributions.* SW designed the analytical inversion system, run the FLEXPART simulations, performed the scientific analysis presented in the paper. AB and IP provided scientific, technical expertise and contributed to the scientific analysis. MS provided the $CH_4$ fluxes from GCP and JDP contributed scientific expertise.

*Competing interests.* The authors have no competing interests.

*Acknowledgements.* We would like to thank all PIs and supporting staff for deploying, maintaining and making available data from observation sites around the Arctic.. In particular, we thank our colleagues from the V.E. Zuev Institute of Atmospheric Optics (Tomsk). We thank the data infrastructure of ICOS (Integrated Carbon Observing System; .icos-cp.eu), the WDCGG (World Data Center for Greenhouse Gases; https://gaw.kishou.go.jp/) and ObsPack (gml.noaa.gov/ccgg/obspack/) for distributing observations. Sophie Wittig has been supported by CEA NUMERICS, funded by the European Union's Horizon 2020 program under Marie Sklodowska-Curie Grant Agreement No 800945.

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
