# Peer review of "Surface networks in the Arctic may miss a future "methane bomb""

_EGUsphere, 2023_

## Referee Comment (RC1)

Review on Wittig et al., egusphere-2023-2308

**General comments**
This study examines whether current observation network is capable for detecting future potential changes in $CH_4$ emissions in the Arctic. Arctic is important as vast amount of carbon is stored and could be released as Arctic warming proceeds, leading to positive climate feedback and enhance global warming. The authors use FLEXPART to generate synthetic observations (using current knowledge of fluxes and meteorological data) and examine whether those data can be used to detect scenario emission changes by an analytical inverse model. Inverse modelling has been widely used to quantify current and history of greenhouse gas budgets, but this study attempts to implement it also for studying future changes. This is a novelty.

The study challenges important questions in climate change, but I have few doubts and questions regarding their choice of methods. Particularly,

- The authors study the Arctic $CH_4$ emission changes in 35 years – this is rather short considering the processes of e.g. permafrost thaw, and in comparison to other scenario studies (which are often up to 2100). Because of the relatively short study period, the $CH_4$ emissions are needed to be increased unrealistically fast (20 % $yr^{-1}$), as authors point out as well, and therefore, the credibility of the results are weak. The choice of length and the increasing rate of emissions need to be justified, and at least add implications for more realistic changes.

- The above point leads to a conceptual question about "methane bomb". In Introduction (P2 L7 – P2 L16), you use this term for both gradual and sudden methane release from the Arctic. As I understood, this study is about the gradual and continuous changes, and this needs to be clarified (Abstract, Introduction and Method).

- If I understood correct, you have generated synthetic data based on present/past prior emission information and meteorological data, which are used to constrain the future scenario fluxes. This would mean that observations would try to adjust emissions to current emission level. So the "detection limit" is when the observations cannot anymore constrain the fluxes to current emission level (+ uncertainty limit), i.e. the limit where observations cannot "see". Is this correct, and what you aim to do? I would assume that it would be more meaningful if you generate synthetic mixing ratio data based on future emission scenarios, and constrain some prior fluxes with that data. With this, you could see if we can detect emission changes even if there are "missing information" in prior fluxes.

- The authors have examined the current and "extended" observation network, but due to the effect of the Russian war, substantial number of surface stations lack of data at current. How likely that we can still detect future changes in $CH_4$ emissions in Eurasia? How long of data lack is critical? I think these are very important questions. You may not need to rerun all simulations without those stations, but adding a few could bring really valuable information about future Arctic $CH_4$ study.

- Following the previous point, you have completely missed about the role of satellite data. I understand that it is challenging to do satellite inversion with Lagrangian models, but I would at least like to see some discussion about it. What if we have had "surface" data at satellite retrieval points?

**Specific comments**

P1 L13-15 Please add references to support your argument. I agree that $CH_4$ emissions from wetlands and other freshwater systems are probably a dominant source, but how large are the other natural sources?

P2 L3-4: Could you add information about how large are the anthropogenic $CH_4$ emissions in the Arctic in comparison to wetland emissions?

At end of Introduction: Please make it clear how many years of future scenarios you study.

P3 L1: Did you optimize the fluxes grid-wise or region-wise (121 sub-regions)?

P6 L7: Could you clarify by "only recently"? What is the year limit you have chosen?

P6 L8: "measurement of $CH_4$ columns" is originally not measuring mixing ratios, but to be used in inversion, you will probably only use the mixing ratio data. Also, satellite data also provide $CH_4$ column information, but those locations are probably not of satellites. This phrase should be clarified better.

Section 3.4:
* Could you possibly change the title to "Generating synthetic $CH_4$ mixing ratio data"?
* What is the temporal resolution of your generated data? 3-hourly?
* Initial concentrations means concentrations in each year (2008–2019)?

P8 L15-18: Please specify a bit more in detail how you have come to 506 different set-ups. It is unclear from the figures/tables as well as from the text. What are the different set-ups, did you change only emission scenarios (as the sentence is is in that section), or did you also use different synthetic data? Did you use different trends, or is all inversions have same trends as presented in Table 2? It is also unclear why there are two similar figures (Figure 2 and 4). Could you possibly combine them?

P9 L8-9: Why "only this region should be updated by the inversion"? Is East Eurasia strictly uncorrelated with other regions? Did you strictly set it so that observations are only constraining this region? If not, it is not surprising that other regions are also affected.

P9 L10: Is it so that the posterior emissions are much lower than the truth because the observations are generated using present-day emissions?

P10 L5-7: Is it really so that the "increase in the simulated scenarios is underestimated"? I wonder how strong are the regional correlations. Also, do you trust the "truth" or posterior estimates? You need to re-think how to put your arguments.

P11 L2: By "combine", do you mean that you only show the results of the region where you modified the trends, i.e. the effect of other regions are not presented? Please make it clearer.

P11 L 20-23: I am not sure what you wish to say. The applied trend is unrealistic, and you hoped that the inversion would detect the changes much earlier? Or you think that you should have applied a bit more realistic trend? What you mean by "more illustrative" – more, compared to what?

P12 L10-11: Is it really true that there is no influence about observations? What if there is a station over there? I would also guess that the observations in surrounding regions could affect the results.

P12 L13-15: This is interesting, but could be also due to the fact that many of the extended stations are often close to the currently available stations. Also, the emission magnitude near the station is important to consider – if we add stations where emissions are small, the effect could be minor.

P13 L1-2: Is there anything you could do to attribute those discrepancies to fluxes by changing some setups/uncertainties? Despite the minor effect on your results, do you still think those sites are important and could bring information about changes in trends in northern Europe or surrounding regions?

**Technical corrections**
Please use same terms for generated mixing ratio data (modelled, generated, synthetic, etc..)

Please check the spaces between units, and follow the journal role.

P1 L10 Remove "temperature"

P5 L10 Section Inversion framework
Please add section number

P11 L10: annual posterior emissions in year j and region r emis aj,r

P11 L12: Please move the $j$ and $r$ ranges on the right hand side of the equation, i.e.
$$\text{emiss}^a - \text{emiss}^b < e, \quad j \in [2021, 2055], r \in [1, 121]$$
You could put "is not fulfilled" in L10. Please also do so in Eq. 5 and 6.

P11 L16: "the threshold year is generally higher"
Do you perhaps mean "the year is generally **later**"?

P11 L25: "terms of detection limits, an increase of a few, up to 10 Tgy -1 , is necessary for statistically reliable detection."
Could you add e.g. in brackets how much they are in percentage?

P15 L18: "TROPOMI CH4" $\rightarrow$ $CH_4$ with subscript.

Figure captions: Use (a), (b) instead of "left" "right".

Figure 3 caption: I feel it would be more appropriate to say e.g. "Location of the sites where synthetic mixing ratio data are generated from", as you do not use actual observations at all.

Figure 5 y-axis: Are those units really correct? For example, in the bottom panel, 100 Tg/month of $CH_4$ from Arctic in 2020 does not sound at all realistic (even if it was annual emission). Y-axis label and caption does not have same units.

Figure 8:
- Please use more informative label in the color bars.
- The unit in color bar is [%], and color scales ranges between $-10^3$ to $10^3$, i.e. 1000% change in emissions. Is this correct?
- Caption for (b): "Difference between the…" $\rightarrow$ "Absolute differences between..""?

---

## Author Comment (AC1)

**Review 1**

The replies to the comments are highlighted in green.

**General comments**

This study examines whether current observation network is capable for detecting future potential changes in CH4 emissions in the Arctic. Arctic is important as vast amount of carbon is stored and could be released as Arctic warming proceeds, leading to positive climate feedback and enhance global warming. The authors use FLEXPART to generate synthetic observations (using current knowledge of fluxes and meteorological data) and examine whether those data can be used to detect scenario emission changes by an analytical inverse model. Inverse modelling has been widely used to quantify current and history of greenhouse gas budgets, but this study attempts to implement it also for studying future changes. This is a novelty.

The study challenges important questions in climate change, but I have few doubts and questions regarding their choice of methods. Particularly,

- The authors study the Arctic CH4 emission changes in 35 years – this is rather short considering the processes of e.g. permafrost thaw, and in comparison to other scenario studies (which are often up to 2100). Because of the relatively short study period, the CH4 emissions are needed to be increased unrealistically fast (20 % yr-1), as authors point out as well, and therefore, the credibility of the results are weak. The choice of length and the increasing rate of emissions need to be justified, and at least add implications for more realistic changes.

The trend of 20% increase per year presented in the study is indeed very drastic. We have chosen this increase to represent a true "methane bomb" event, where large amounts of $CH_4$ are released in a relatively short period of time. Other, possibly more realistic trends, starting with an increase of 0.1% per year, were indeed simulated as part of this experiment. However, as similar results in terms of network detectability were obtained with smaller trends, we have chosen to present only the results from the highest trend scenario in this paper, as it shows that there is a problem with the ability of the networks to detect even very large changes.

We propose to add the following in section 3.5 to clarify:

"For each of these zones, positive trends are applied separately on wetland and anthropogenic emissions. Oceanic $CH_4$ emissions are only increased in the sub-region that contains the ESAS, as these are difficult to detect with the surface networks.

The trends are hereby varied between a 0.1 and 20% increase per year for anthropogenic and wetland emissions and between 1 and 100% for oceanic sources. As the results obtained are applicable to both lower and higher trend scenarios, we focus only on the highest selected increase (20% for wetlands and anthropogenic sources and 100% for oceanic fluxes), as this is also the most representative for a "methane bomb" event."

- The above point leads to a conceptual question about "methane bomb". In Introduction (P2 L7 – P2 L16), you use this term for both gradual and sudden methane release from the Arctic. As I understood, this study is about the gradual and continuous changes, and this needs to be clarified (Abstract, Introduction and Method).

In literature, the term "methane bomb" has indeed been used for both steady and sudden increase in $CH_4$ emissions. In this study we define a "methane bomb" event as a sudden and steep increase in methane emissions releasing large amounts of $CH_4$ over a few years. A similar definition has e.g. been used by Schuur et al. (2022).
We added clarifications of this term in the introduction and abstract.

- If I understood correct, you have generated synthetic data based on present/past prior emission information and meteorological data, which are used to constrain the future scenario fluxes. This would mean that observations would try to adjust emissions to current emission level. So the "detection limit" is when the observations cannot anymore constrain the fluxes to current emission level (+ uncertainty limit), i.e. the limit where observations cannot "see". Is this correct, and what you aim to do? I would assume that it would be more meaningful if you generate synthetic mixing ratio data based on future emission scenarios, and constrain some prior fluxes with that data. With this, you could see if we can detect emission changes even if there are "missing information" in prior fluxes.

The generated synthetic data is indeed based on present and past prior emissions data, however, as described in the manuscript, we apply trends on these current emission scenarios to generate possible future emission scenarios which we define as the *truth*, thus generating a positive trend on future observations. The prior state of the fluxes is given by current flux estimates and remains unchanged throughout the whole period under study since this is the best guess that we have for the actual $CH_4$ fluxes in the Arctic region. Ideally, the synthetic observations should move these prior flux estimates to the "true" flux estimates, which is not the case: the prior fluxes are not brought close enough to the true fluxes by assimilating data representative of the truth. This is observable even in the first years of the period under study, where the true state is well within the uncertainty ranges of the prior emissions. Therefore we conclude that the problem lies within the limitations of the practical implementation of data assimilation (i.e., in this case, the sparseness of the network).

- The authors have examined the current and "extended" observation network, but due to the effect of the Russian war, substantial number of surface stations lack of data at current. How likely that we can still detect future changes in CH4 emissions in Eurasia? How long of data lack is critical? I think these are very important questions. You may not need to rerun all simulations without those stations, but adding a few could bring really valuable information about future Arctic CH4 study.

The lack of availability of measurement data as a result of the current war is undoubtedly an important obstacle. When we first implemented this study, this event had only just begun and

the assumption at the time was that the war and the associated restrictions would hopefully not last long. Nevertheless, we believe it is important to carry out our study outside of the current political situation, as this may change at any time.

Additionally, the data of the Russian measurement stations may not be available for many European institutions, however other countries did not interrupt their collaboration with the Russian scientific community. Insights in the methane emissions may therefore still be provided in studies carried out by scientific institutions not affected by current sanctions. Apart from that, the current measurement network in Russia is already relatively sparse, and $CH_4$ fluxes can only be constrained to a limited extent and in certain regions. Excluding these stations would undoubtedly remove the limited insight that we have into the Russian Arctic, even assuming that many stations could be deployed all around Russia, which can be concluded even without implementing new simulations.

We propose to clarify in section 3.2 as well as in the conclusion:

"Both the current and extended networks were selected based on their theoretical provision of $CH_4$ measurements, including measurements in the Russian Arctic that may not currently be accessible to the scientific communities of certain countries, as we believe it is important to conduct this work outside of ongoing political conflicts."

"Current political differences as well as the associated sanctions are an additional obstacle regarding the accessibility of crucial $CH_4$ observations in the Russian Arctic and Sub-Arctic. As the network in this region is already limited, this missing information may further hamper obtaining a complete picture of ongoing processes in the Arctic, including the detection of a possible methane bomb."

- Following the previous point, you have completely missed about the role of satellite data. I understand that it is challenging to do satellite inversion with Lagrangian models, but I would at least like to see some discussion about it. What if we have had "surface" data at satellite retrieval points?

Satellite observations have indeed a high potential to compensate for the lack of stationary observation sites in the Arctic in the future and studying an "ideal" scenario of available satellite data in high northern latitudes would undoubtedly be valuable. However, in this study, we focus on the stationary observation network since currently operating remote sensing instruments still face technical obstacles in high northern latitudes. Future satellite missions may offer a better coverage of data in the Arctic, however, these observations will not be available for several years and the final quality of these data cannot yet be predicted.

We propose to add to Section 3.2:

"In this study, we focus exclusively on stationary $CH_4$ measurements, as our period of study spans several decades. Other types of greenhouse gas measurements, such as satellite observations, are currently limited to providing data for only a few years and are therefore not suitable for our purposes."

And modify the following paragraph in the conclusion:

"Therefore, efforts to integrate mobile campaigns and new-generation satellite observations into inverse modelling systems should be supported and developed further. Satellite observations in particular offer a high potential to compensate for the lack of in situ observations in the Arctic. The feasibility of using available satellite data products for inverse modelling of methane emissions in high northern latitudes was, for instance, discussed by Berchet et al., 2015 and several approaches integrating these observations in Arctic regions (e.g., TROPOMI $CH_4$ products, Tsuruta et al., 2023) have been implemented. However, the quality of the data provided by currently operating remote sensing instruments is hampered in high northern latitudes by factors such as high solar zenith angles, low albedo of the Arctic Ocean and limited daylight during polar nights.

However, new satellite missions (e.g., the Franco-German MERLIN project) will possibly provide large, accurate and high-resolution data sets, suitable for characterising $CH_4$ plumes from regional sources and better constraining methane fluxes in the Arctic."

**Specific comments**

P1 L13-15 Please add references to support your argument. I agree that CH4 emissions from wetlands and other freshwater systems are probably a dominant source, but how large are the other natural sources?

References have been added to support the argument and estimates added.

P2 L3-4: Could you add information about how large are the anthropogenic CH4 emissions in the Arctic in comparison to wetland emissions? At end of Introduction: Please make it clear how many years of future scenarios you study.

Estimates as well as years were added in the introduction.

P3 L1: Did you optimize the fluxes grid-wise or region-wise (121 sub-regions)?

The fluxes are optimised region-wise. Clarification has been added.

P6 L7: Could you clarify by "only recently"? What is the year limit you have chosen?

Since 2022. Clarification has been added.

P6 L8: "measurement of CH4 columns" is originally not measuring mixing ratios, but to be used in inversion, you will probably only use the mixing ratio data. Also, satellite data also provide CH4 column information, but those locations are probably not of satellites. This phrase should be clarified better.

We propose to rephrase:

"the stations use ground-based remote sensing instruments to obtain total column measurements $CH_4$"

Section 3.4:
• Could you possibly change the title to "Generating synthetic CH4 mixing ratio data"?

The title has been adjusted.

• What is the temporal resolution of your generated data? 3-hourly?

In this study, we only generated monthly measurements in order to limit the size of the observation vector.

• Initial concentrations means concentrations in each year (2008–2019)?

Yes, the initial concentrations from the years 2008 to 2019 were used to obtain the background mixing ratios.

P8 L15-18: Please specify a bit more in detail how you have come to 506 different set-ups. It is unclear from the figures/tables as well as from the text. What are the different set-ups, did you change only emission scenarios (as the sentence is is in that section), or did you also use different synthetic data? Did you use different trends, or is all inversions have same trends as presented in Table 2? It is also unclear why there are two similar figures (Figure 2 and 4). Could you possibly combine them?

Wetland and anthropogenic sources were increased in each of the 121 sub-regions as well as the 5 supra-regions, which gives 126 regions in total. We use 2 different observation networks. That gives 252 (126 x 2) scenarios for wetlands and 252 scenarios for anthropogenic sources. Oceanic sources were only increased in 1 region, using 2 different networks, which gives 2 scenarios. So in total there are 506 (252 + 252 + 2) scenarios. A more extensive description has been added to the section.

P9 L8-9: Why "only this region should be updated by the inversion"? Is East Eurasia strictly uncorrelated with other regions? Did you strictly set it so that observations are only constraining this region? If not, it is not surprising that other regions are also affected.

Only this region should be updated because only this region is truly higher than in the prior, which should, ideally, be seen by the inversion. However, this is not the case, since the data assimilation is not perfect in our study,

P9 L10: Is it so that the posterior emissions are much lower than the truth because the observations are generated using present-day emissions?

The observations are in fact *not* generated using present flux estimates but using future emission estimates. This is explained in Section 2 of the manuscript. (See also answer to third comment of general comments.)

P10 L5-7: Is it really so that the "increase in the simulated scenarios is underestimated"? I wonder how strong are the regional correlations. Also, do you trust the "truth" or posterior estimates? You need to re-think how to put your arguments.

In our inversion set-up, we do trust the "truth" over the posterior state - since we define it as the true, future state of the fluxes. Which does not mean that these emissions are likely to occur in reality, but they are the true state of the inversion set-up and thus, the posterior state should be optimised accordingly.
We propose to clarify:

"This means that the increase retrieved in the posterior state is underestimated compared to the generated truth in the "correct" area, which is considered to be the true state of the emissions in this inversion set-up. This is partially compensated for in the total posterior by overestimations in the same emission sector in different regions."

P11 L2: By "combine", do you mean that you only show the results of the region where you modified the trends, i.e. the effect of other regions are not presented? Please make it clearer.

Yes, in the figures in section 4.2 we present 121 set-ups with elevated trends in each corresponding region. The effect on other regions from each scenario has been evaluated with the corresponding equation in section 4.2.3. We considered this the best way to present the variety of results of the different configurations. The introduction to Section 4.2 has been extended, more detailed descriptions of the maps and the corresponding calculations are in the sub-sub-sections.

P11 L 20-23: I am not sure what you wish to say. The applied trend is unrealistic, and you hoped that the inversion would detect the changes much earlier? Or you think that you should have applied a bit more realistic trend? What you mean by "more illustrative" – more, compared to what?

Here, we want to express that it's not so much the year of detection which is interesting, since it's based on a probably unrealistically fast methane bomb, but the smallest amount of emissions which can be detected, whenever they may happen in time.
We propose to clarify:

"Hence, it is more illustrative to analyse the smallest amount of emissions which can be detected, as shown in Figure 6b, than simply using the year of detection as an indicator. "

P12 L10-11: Is it really true that there is no influence about observations? What if there is a station over there? I would also guess that the observations in surrounding regions could affect the results.

In the oceanic regions that we describe, $CH_4$ emissions from wetlands are 0 in the emission data set (which is logical since wetland emissions do not occur in the Ocean). Hence, our generated *truth* is also 0 as well as the computed posterior emissions. That observation sites in these oceanic regions would further improve the detection is unlikely - since in our set-up there are no emissions to be detected in the first place.
As for the neighbouring regions, in data assimilation, any additional data is generally beneficial. So in a sense having observation sites in the Ocean could potentially bring information on wetland emissions, if the oceanic sites are sampling air masses coming from

wetland regions. The interest of these data compared to the large uncertainties in the transport and the difficulty of maintaining such oceanic sites is however questionable.

P12 L13-15: This is interesting, but could be also due to the fact that many of the extended stations are often close to the currently available stations. Also, the emission magnitude near the station is important to consider – if we add stations where emissions are small, the effect could be minor.

This is certainly true. However, in remote regions it is difficult to locate hypothetical observation sites, as they may be in regions where both construction and maintenance are not feasible. Therefore, for this study, we have chosen actual, potential observation sites where we know it is technically possible to measure $CH_4$ mixing ratios.
We propose to add:

"One possible reason for this could be related to the locations of the additional observation sites, as several of them are located close to operating measurement stations and/or in areas with low estimated $CH_4$ fluxes."

P13 L1-2: Is there anything you could do to attribute those discrepancies to fluxes by changing some setups/uncertainties? Despite the minor effect on your results, do you still think those sites are important and could bring information about changes in trends in northern Europe or surrounding regions?

These observation sites may not be beneficial in detecting a potential methane bomb in the Arctic, which was the focus in our study, but they are certainly useful for other cases, e.g. European scale monitoring of $CH_4$ emissions.

**Technical corrections**

All technical corrections have been implemented, unless there were justified objections from our side not to make the changes (see responses to comments).

Please use same terms for generated mixing ratio data (modelled, generated, synthetic, etc..)

Please check the spaces between units, and follow the journal role.

P1 L10 Remove "temperature"

P5 L10 Section Inversion framework
Please add section number

P11 L10: annual posterior emissions in year j and region r emis aj,r

P11 L12: Please move the j and r ranges on the right hand side of the equation, i.e. emissa – emissb < e, j∈[2021, 2055], r∈[1, 121] You could put "is not fulfilled" in L10. Please also do so in Eq. 5 and 6.

We prefer to have the range of $j$ and $r$ below the equations, as this is a legitimate way to define them and there are no contradicting guidelines from the journal.

P11 L16: "the threshold year is generally higher" Do you perhaps mean "the year is generally later"?

P11 L25: "terms of detection limits, an increase of a few, up to 10 Tgy -1 , is necessary for statistically reliable detection." Could you add e.g. in brackets how much they are in percentage?

P15 L18: "TROPOMI CH4" → CH4 with subscript.

Figure captions: Use (a), (b) instead of "left" "right".

Figure 3 caption: I feel it would be more appropriate to say e.g. "Location of the sites where synthetic mixing ratio data are generated from", as you do not use actual observations at all.

Figure 5 y-axis: Are those units really correct? For example, in the bottom panel, 100 Tg/month of CH4 from Arctic in 2020 does not sound at all realistic (even if it was annual emission). Y-axis label and caption does not have same units.

The prior annual emissions are actually in this magnitude for the "*entire region*" (which encompasses a larger area than the Arctic) shown in the supplements. This is predominantly due to the wetland emission data set we used as prior information (Poulter et al., 2017) which makes up around 50% of those emissions.
Y-label has been adjusted.

Figure 8:
• Please use more informative label in the color bars.
• The unit in color bar is [%], and color scales ranges between -103 to 103, i.e. 1000% change
in emissions. Is this correct?
• Caption for (b): "Difference between the…" → "Absolute differences between.."?

---

## Author Comment (AC2)

**Review 2**

The replies to the comments are highlighted in green.

The authors present a study that aims at assessing what enhancement level of Arctic CH4 emissions may be reliably detected, and spatially attributed, based on GHG mixing ratio observations from the pan-Arctic tall tower network. Their approach uses atmospheric transport modeling with FLEXPART to first general synthetic time series of mixing ratios that reflect the changes in the atmosphere following surface flux enhancements at selected regions. In subsequent steps, these synthetic time series are then used as input for atmospheric inversions in an attempt to quantify, and spatially attribute, the surface flux rates of CH4 corresponding to the chosen emission scenarios. Since in this synthetic setup the 'truth' is known, this approach allows to quantify how well the inversion-based posterior fluxes agree with the true emissions, at what level of flux enhancement the higher fluxes are significantly different from the baseline, and how well the flux trends are assigned to the correct target regions. Based on these metrics, the authors conclude that substantial flux enhancements are required for a reliable detection, particularly in regions with sparse observations, and that a mis-attribution of the flux signals is quite common.

The overall objectives that Wittig et al. aim at are highly relevant – even in the absence of a 'methane bomb' scenario, enhanced GHG emission rates from degrading Arctic permafrost can be expected under future climate change, and a monitoring system that would reliably pick such changes would therefore be very useful. The in-situ atmospheric GHG monitoring tower network, in combination with atmospheric inverse modeling, is a suitable tool for this purpose, but due to the sparse network coverage the sensitivity of this tool towards future changes is still uncertain. The approach used within the context of this study, i.e. generating synthetic datasets with a known truth that allows to assess how well trends are quantified, and how reliable the spatial attribution of fluxes is, is well suited for this purpose. Unfortunately, some settings in the inversion setup seem to be over-simplified, so that even though the qualitative results may be solid, most quantifications are very questionable.

I see 3 major issues that compromise the findings presented in this work:

First, the authors produce synthetic mixing ratio observations as a prerequisite for conducting inversions for future emission scenarios, but they do not apply uncertainties when using this information in the actual inversions. Or rather, such uncertainties are not described in the presented paper, but based on the statement on p.11, ll.17-18 (These figures reflect an ideal case where uncertainties in the inversion system are minimized) I presume that none were applied. If this is the case, then the same transfer functions were applied in forward (to produce the synthetic data) and in backward modes (to execute the inversion). This is an over-simplification of the situation in a regular atmospheric inversion, where model-data mismatch uncertainties such as e.g. transport, mixing, or aggregation errors are a key component that make the links between surface processes and atmospheric observations much more challenging. Accordingly, all quantitative findings presented herein, such as years until detection of a change, or emission thresholds until detection, are highly questionable, and detection limits are likely low-biased.

Indeed, we do not apply uncertainties on the synthetically generated observation (in forward mode). We are aware of this aspect and chose this method purposefully. The aim of our work is to evaluate the current stationary observation network in the Arctic region with regard to the detection of a possible methane bomb. Our main conclusion is that the network is not fully adequate for this purpose and is limited in its ability to detect small changes in emissions in the Arctic. Introducing uncertainties on the observations would lead to a similar conclusion, but with an even higher detection threshold.

We propose to clarify in Section 2:

"Theoretically, the synthetic observations $y$ should be perturbed by an error $\in$ (with a Gaussian distribution, following the matrix $R$), accounting for measurement errors, as well as other uncertainties such as transport and aggregation (described e.g. by Szénási et al., 2021). In our approach, we deliberately disregard these errors in order to obtain optimistic results and assimilate optimal measurements to analyse the best possible detection of different observation networks (Section 3.2) regarding a methane bomb event."

Additionally, we propose to add a paragraph summarising our general approach including clarification on the synthetic observations:

"In order to implement this work, we apply hypothetical trend scenarios on different $CH_4$ emission sources to simulate a methane bomb in different regions located in high northern latitudes. By combining these emission scenarios with the extrapolated output of an atmospheric transport model, we obtain synthetic $CH_4$ mixing ratios for the current observation network in the Arctic and Sub-Arctic as well as for an observation network extended by possible additional sites. These synthetic observations subsequently serve as input data for the inverse modelling setup in order to identify a temporal threshold of possible detection and to analyse regional differences in the ability of the two networks to adequately detect and localise increasing $CH_4$ emissions. Since we assume optimum quality and availability of the measurement data, the results obtained represent a best-case scenario for the detection of an Arctic methane bomb using exclusively in situ observations."

As well as adding to the conclusion:

"In this approach, we have made the optimistic assumption of excellent quality and availability of measurement data. The results presented therefore represent the best possible scenario for detecting a future Arctic methane bomb."

Second, the fact that pan-Arctic posterior fluxes are systematically low-biased, compared to the 'true' fluxes in this synthetic experiment, suggest that the chosen setup is compromised. The best explanation I could come up with to interpret this phenomenon is that the correlation length scale chosen for FLEXPART does not allow to reproduce the steep gradients in regional flux patterns that emerge when emissions only in one region are ramped up extremely, while the neighboring regions stay at their low prior values. Such gradients obviously pose a major challenge to any inversion framework fed by sparse observations, where gaps in between monitoring sites need to be interpolated based on assumed spatial relationships within flux fields, necessarily producing (to a varying degree) smooth result surfaces. There may be other factors at play here, but emission peaks within target regions that are systematically underestimate, while adjacent regions have a high bias

in fluxes, may be related to this. In any case, the problem requires further investigation, and in-depth discussion, both of which is currently lacking in this manuscript.

The transport model FLEXPART used in this work does not contain a modifiable function regarding the correlation length. Here, we use the footprints obtained from simulated backward trajectories to determine both the synthetic observations as well as their modelled equivalents, which serve as input for the inversion framework. We would assume that your comment about the correlation length refers to the spatial correlation of the prior error covariance matrix in the inverse modelling set-up, which is 500 km (mentioned in Figure 1). Information on the spatial and temporal correlations have been added in Section 2:

"The off-diagonal elements of the prior error covariance matrix are thereby determined by applying spatial and temporal correlations of 500 km and 7 days, respectively"

For a  more comprehensive description we refer to our previous work (Wittig et al, 2023) to avoid unnecessary repetition.

Furthermore, as described in Sections 3.5, we did not only attribute a steep increase in methane emissions in each of the individual sub-regions separately, we also increased the emissions in much larger regions (see supplements, page 4, Figure S1). As demonstrated in Section 4.1, similar discrepancies between the obtained posterior results and the assumed *true* state of the fluxes were obtained.

In addition to that, our inverse modelling set-up also optimises the background concentrations alongside $CH_4$ fluxes. This has briefly been mentioned in the description of the state vector in Section 2. Since the prior background estimations are not perfect, part of the missing $CH_4$ mass may be compensated by increasing the posterior background concentrations.

We propose to add the following explanation in Section 2:

"In our analysis of the detectability of elevated Arctic $CH_4$ emissions (Section 4), we examine how accurately the truth is  captured in the posterior emissions of different regions and whether these elevated fluxes are localised in the right area. By design, our inverse modelling system will try to fit additional fluxes by adding $CH_4$ emissions in the Arctic region, but possibly not at the correct location. Since, as described above, the background mixing ratios are also included in the control vector $\mathbf{x}^b$ and consequently optimised in the posterior state, part of the missing $CH_4$ mass is likely to be compensated by increasing the background, hence generating a low-bias in the posterior emissions."

We have additionally highlighted the optimization of the background in Sections 2:

"Here, $\mathbf{x}^b$ also contains information on the initial $CH_4$ background mixing ratios (described in Section 3.4), which  are therefore optimised in addition to the $CH_4$ fluxes."

And Section 3.4:

"However, since an exact estimate of the background mixing ratios remains challenging and the calculated background concentrations do not provide perfect estimates, the background mixing ratios are optimised together with the $CH_4$ fluxes (see Section 2)."

Third, I find several issues in the statistical measures used to evaluate flux trends:

- Equation (4) compares prior fluxes   (in 2020) to posterior fluxes (in future years) and relates them to          observational uncertainties to assign a detection limit. This calculation is only valid if prior and posterior fluxes in 2020 are      exactly the same, for every target region. Since this study is based      on synthetic data, there should not be a major adjustment between   prior and posterior in the absence of a trend in emissions; however,   since the observational network is sparse, there is reason to assume          that prior and posterior are not identical even in 2020, which may lead to systematic difference even when aggregating fluxes by      region. Without demonstrating that priors and posteriors are          identical on a regional basis, the posterior fluxes in 2020 should         be used as a reference (emis_b_2020), instead of the priors.

  The prior fluxes in 2020 are indeed identical to the truth. The first year in which the true emissions deviate from the prior emissions is therefore 2021. This has been shown in equation 3 (Section 2). To further clarify this fact, we propose to add in Section 2:

  "This trend was only applied from the second year of the study period (2021), in the year 2020 the truth is identical to the prior state."

- Equation (5) compares 'true'  to 'posterior' fluxes to quantify how much of the emission          enhancement is actually captured by the inverse model. This is called 'detected trend magnitudes' in the section header, so the      intention is obviously to quantify how much of the trend is          captured. However, equation (5) compares absolute fluxes, not flux    enhancements since 2020. For quantifying how much of the trend is    actually captured by the inversion, I would find it more convincing      to calculate how much the 'true' flux changed since 2020, and how much of a change is seen between the posterior fluxes over the          same time span.

  In this analysis, we want to determine how well the *true* trend (which is known in our case since we determined it) is detected in each region by the inversion, not which total trend each region shows since 2020, as this would not give us any information about the performance of the observation network. Therefore, the deviation of the posterior state in the threshold year from the truth is a good indicator, since, if the trend would be adequately detected, the difference between the posterior emissions and the truth would be minor.

  For clarification, we propose to state at the beginning of 4.2.2:

"Subsequently, we want to examine how well the previously determined trends of 20% increase in wetland emissions and 100% increase in oceanic $CH_4$ emissions, respectively, are captured in each of the corresponding sub-regions."

In addition to that, we propose to extend our analysis to determine how much of the increase in $CH_4$ fluxes is detected over the whole Arctic domain by the inversion when increasing the $CH_4$ emissions in one of the sub-regions (see following comment). We therefore extended Figure 7 by two subfigures (now Figure 6c and 6d in the edited manuscript) and added the following description:

"Additionally, in order to determine the share of the truth detected by the inversion, we calculate the detection ratio $K_{j,r}$. Hereby, the posterior increment in all regions $\Sigma\Delta emisa_{j,r}$ in the threshold year j is divided by the the true increment $\Delta emis^t_{j,r}$ in region r:

$$K_{j,r} = \sum \Delta emis^a_{j,r} / \Delta emis^t_{j,r}$$

with $j \in [2021, 2055]$ and $r \in [1, 121]$. Hence, we analyse how much of the true increase is detected, independent from the location it is attributed to, when increasing the $CH_4$ emissions in one of the sub-regions. Higher values indicate that a larger share of the true emissions is detected in the posterior emission, distributed over the whole pan-Arctic domain. Figure 6c shows that the detection ratio is generally higher when the true emissions are increased in regions with a dense observation network (such as North America), with values of up to 100 %. Similar to the relative difference (Figure 6a), the high detection ratios in the oceanic regions are due to the absence of trends in the true emissions, since the $CH_4$ emissions in these regions are nearly zero [...] Regarding the comparison of the detection ratio of the two networks, shown in Figure 6d, the improvement is even smaller with a maximum of 0.3 %."

[Figure]

(a)                    (b)

(c)                    (d)

- I cannot really follow the logic behind equation (6), even    though the objective to quantify mis-attribution is highly relevant.    Why is the 'delta_emis' measure used here? The authors state    themselves in Section 4.2.2 that a good 'delta_emis' factor, i.e. with a value close to zero, indicates that the posterior is    very close to the true emissions. Now when dividing the sum of    'delta_emis' in other regions by the 'delta_emis' from the study region, if the latter value is close to zero the result would    be rather high increment ratios ..?? So why use the 'delta' measure in the first place here? I would find it much more intuitive if the authors first quantify how much integrated pan-Arctic flux    budgets were increased when raising fluxes in a single region, i.e.   how much of that 'true' signal is actually detected by the    inversion, no matter where exactly. Next, you should simply quantify    what fraction of that enhancement is attributed to the target region where fluxes were enhanced, and what fraction lies outside.

The parameter $\Delta emis^a_{j,r}$ used in equation 6 is not equal to $\Delta emis_{j,r}$ calculated in equation 5 (hence the use of the exponent "a"). It is rather the increment of the posterior fluxes in the threshold year since the year 2020. We have edited the description of this parameter to avoid misunderstandings:

"$\Delta emis^a_{j,r}$ and $\Delta emis^a_{j,i}$ hereby represent the difference between the posterior $CH_4$ emissions in the threshold year j and the true emissions in the year 2020 in the corresponding region r or i, respectively."

Thus, we calculate the sum of the posterior increments in all other regions divided by the posterior increment of the targeted sub-region. This value should ideally be zero, since this would indicate that no posterior increment would be detected outside the region in which the fluxes were increased. Higher absolute values on the other hand indicate that the increment outside the target region is higher than in the region itself.

Your suggestion to quantify the integrated pan-Arctic flux budget when raising fluxes in one of the sub-regions was implemented in Section 4.2.2 (see previous comment).

Moreover, we propose to extent Figure 8 by two subfigures (now figures 7c and 7d in the edited manuscript) with the following analysis in order to determine the misattribution of the true fluxes:

"In addition to the posterior increment ratio, we compute the true increment ratio $\kappa^t_{j,i}$ for each sub-region i:

$\kappa^t_{j,i} = \sum \Delta emis^a_{j,r} / \Delta emis^t_{j,i}$

for the threshold year j ∈ [2021, 2055] and the region r ∈ [1, 121] r ≠ i. $\Delta emis^t_{j,i}$ is hereby defined as the difference between the true $CH_4$ fluxes in the threshold year j and the truth in 2020 in the corresponding region i. The closer the value of $\kappa^t_{j,i}$ of a specific region is to zero, the less true emissions are misattributed to other sub-regions. The true increment ratios are shown in Figure 7c. Similar to the posterior increment ratios, the fluxes are generally less misattributed when the true

emissions are increased in continental areas with available observation sites, especially in Siberia and Canada. The improvements from the extended observation network are smaller regarding the true increment ratio (see Figure 7d) in comparison to the posterior increment ratio, with only one region in eastern Siberia showing a clear improvement of around 10%."

[Figure]

**Additional comments:**

- I don't find the flow charts      (Figs. 1, 4) too helpful in the current format.

  Figure 1 and 4 have been edited. Figure 1 has been included in the supplements instead and additional description has been added to Figure 4.

- Using emissions, or emission            thresholds, in absolute numbers (e.g. Tg/yr) is misleading, since        the size of the regions is variable, and unknown to the reader.          Fluxes would be more intuitive if normalized by area.

  For our purposes, we consider it relevant to have absolute numbers for the emission threshold. Even though the sub-regions have varying sizes, we are more interested in the "burden" on the atmospheric $CH_4$ concentrations by a given region.

- When presenting the 'inversion          method' as Section 2, some details are missing. Maybe it would be      better to place this after the 'material' section.

  We would prefer to leave the methods before the material, as we repeatedly refer to the content of the inverse modelling section in the material section. We think it is difficult to understand why, for example, the synthetic observations are created

without knowing the background of the inverse modelling set-up. Additional references to the material sections have been added in the methods section for clarification.

- In Section 3.2, definitions are          not 'clean', since the assumption of continuous data everywhere       already upgrades the 'current' network. Also, some more details on the networks, e.g. total number of sites, or regional        distribution, should be added to the text.

We propose to clarify and add:

"The term "current" refers hereby only to the location of the stations. This network, as used in this study, already provides additional data compared to the actual observations available from these sites. This is because, as stated before, we assume continuous measurements where currently only flask measurements are carried out. The current network contains hereby 40 stations in total, whereby the majority (26 sites) of the sites is located in North America (Canada, USA and Greenland). 10 observation sites are located in the Russian Arctic and Sub-Arctic and 4 sites in Northern and Western Europe (Finland, Norway, Ireland and Iceland). [...] The extended network expands the current network by 16 observation sites. The majority of these stations, 11, are located in Northern Europe (Sweden, Finland, Norway, Lithuania and East Russia), 3 in Central and Western Russia and one station each in Canada and Greenland."

- In Section 3.4, more information on the setup of FLEXPART and the optimization strategy would be helpful.

The configurations used for FLEXPART have been described as detailed as possible in the current manuscript (page 7, lines 7 to 12). There are no other parameters in this transport model that are relevant for the inversion and that we could specify further. For more detailed information on the function of FLEXPART version 10.3, we refer to Pisso et al., 2019.

Additional information on how the FLEXPART simulations have been used as input for the inverse modelling framework have been added in Section 3.4:

"The so-called footprints obtained by sampling the near-surface residence time of the various backward trajectories of the virtual particles are subsequently used to determine the $CH_4$ mixing ratios per methane emission sector (Section 3.3) and sub-region (Section 3.1). The footprints define hereby the connection between the methane fluxes discretised in space and time and the change in concentrations at the observation site (Seibert and Frank, 2004). To obtain a time series of modelled $CH_4$ mixing ratios, a time series of footprints is integrated with discretised $CH_4$ flux estimates. As described in Section 2, in the inverse modelling framework, the modelled $CH_4$ mixing ratios obtained from the FLEXPART footprints are included in the observation operator **H**. In this study, this matrix is used for both the calculation of the synthetic future observations (shown in Equation 3) based on future emission

scenarios (see Section 3.5) as well as their modelled equivalents based on prior emission estimates."

The optimization strategy is an analytical Bayesian inversion framework (equation 1 and 2) as described in Sections 2. This framework, including the components as well as the corresponding input data, have been extensively described in the manuscript and the references provided here cover the main points.